# Skeletal Muscle-Derived Stem Cell Transplantation Accelerates the Recovery of Peripheral Nerve Gap Injury under 50% and 100% Allogeneic Compatibility with the Swine Leucocyte Antigen

**DOI:** 10.3390/biom14080939

**Published:** 2024-08-02

**Authors:** Tetsuro Tamaki, Toshiharu Natsume, Akira Katoh, Atsuko Shigenari, Takashi Shiina, Nobuyuki Nakajima, Kosuke Saito, Tsuyoshi Fukuzawa, Masayoshi Otake, Satoko Enya, Akihisa Kangawa, Takeshi Imai, Miyu Tamaki, Yoshiyasu Uchiyama

**Affiliations:** 1Muscle Physiology and Cell Biology Unit, Tokai University School of Medicine, 143 Shimokasuya, Isehara 259-1193, Kanagawa, Japan; natsumetoshiharu@gmail.com (T.N.); akiraka@tokai.ac.jp (A.K.); nn3226@tokai.ac.jp (N.N.); kousukesaitou427@yahoo.co.jp (K.S.); ft8772@tokai.ac.jp (T.F.); tokai-takeshi@tokai.ac.jp (T.I.); miyu.tamaki90@gmail.com (M.T.); y-uchi@tokai.ac.jp (Y.U.); 2Department of Physiology, Tokai University School of Medicine, 143 Shimokasuya, Isehara 259-1193, Japan; 3Department of Molecular Life Science, Tokai University School of Medicine, 143 Shimokasuya, Isehara 259-1193, Japan; ashige@tokai.ac.jp (A.S.); tshiina@tokai.ac.jp (T.S.); 4Department of Urology, Tokai University School of Medicine, 143 Shimokasuya, Isehara 259-1193, Japan; 5Department of Otolaryngology, Tokai University School of Medicine, 143 Shimokasuya, Isehara 259-1193, Japan; 6Department of Radiation Oncology, Tokai University School of Medicine, 143 Shimokasuya, Isehara 259-1193, Japan; 7Swine and Poultry Research Center, Shizuoka Prefectural Research Institute of Animal Industry, Kikugawa 439-0037, Shizuoka, Japan; micropig@sp-exp.pref.shizuoka.jp (M.O.); satoko1_enya@pref.shizuoka.lg.jp (S.E.); iza.dai2syougai.1koshide@gmail.com (A.K.); 8Department of Orthopedic Surgery, Tokai University School of Medicine, 143 Shimokasuya, Isehara 259-1193, Japan

**Keywords:** preclinical large animal experiment, micro-mini pig, cloning, allogeneic cellular therapy, swine leukocyte antigen haplotypes

## Abstract

Pig skeletal muscle-derived stem cells (SK-MSCs) were transplanted onto the common peroneal nerve with a collagen tube as a preclinical large animal experiment designed to address long nerve gaps. In terms of therapeutic usefulness, a human family case was simulated by adjusting the major histocompatibility complex to 50% and 100% correspondences. Swine leukocyte antigen (SLA) class I haplotypes were analyzed and clarified, as well as cell transplantation. Skeletal muscle-derived CD34+/45− (Sk-34) cells were injected into bridged tubes in two groups (50% and 100%) and with non-cell groups. Therapeutic effects were evaluated using sedentary/general behavior-based functional recovery score, muscle atrophy ratio, and immunohistochemistry. The results indicated that a two-Sk-34-cell-transplantation group showed clearly and significantly favorable functional recovery compared to a non-cell bridging-only group. Supporting functional recovery, the morphological reconstitution of the axons, endoneurium, and perineurium was predominantly evident in the transplanted groups. Thus, Sk-34 cell transplantation is effective for the regeneration of peripheral nerve gap injury. Additionally, 50% and 100% SLA correspondences were therapeutically similar and not problematic, and no adverse reaction was found in the 50% group. Therefore, the immunological response to Sk-MSCs is considered relatively low. The possibility of the Sk-MSC transplantation therapy may extend to the family members beyond the autologous transplantation.

## 1. Introduction

Autologous transplantation of the somatic mesenchymal stem cells is the best way to address all of the donor and recipient problems in regenerative medicine [1,2,3], although the availability/volume of donor tissues/cells is limited [4,5]. Therefore, the necessity of allogeneic transplantation based on the major histocompatibility complex (MHC) or human leukocyte antigen (HLA) has increased. As a result, various somatic stem cells have been identified, such as adipose tissue-derived stem cells [6,7], bone marrow-derived stem cells [8], dental pulp-derived stem cells [9], and skeletal muscle-derived stem cells [10]. To achieve regenerative medicine, their differentiation and therapeutic tissue reconstitution capacities have been extensively studied in rodent experiments over two decades. However, it is likely to admit that the somatic tissue stem cells showed limited potential for cell differentiation and tissue reconstitution [11].

We have identified skeletal muscle-derived stem cells (SK-MSCs) in mice and sorted them into Sk-34 (CD34+/45−) and SK-DN (CD34−/45−/29+) cells directly after enzymatic isolation (before cell culture) [12,13]. Their multipotent differentiation capacity in vivo has been confirmed in peripheral nerve (Schwann cells, perineurium/endoneurium), skeletal muscle, and blood vessel-related (pericytes, endothelial cells, and vascular smooth muscle cells) using immunohistochemistry and immunoelectron microscopy, along with physiological functional assessment under electrical stimulation using mice and human Sk-MSCs [14,15,16]. Recently, the Sk-34 and Sk-DN cells were isolated from pig skeletal muscle and confirmed similar cell differentiation capacities [17]. Thus, their existence has been consistently proven in three species: mice, humans, and pigs [15,17,18].

In the development of stem cell transplantation therapy, large animal experiments are important and inevitable for applying the basic scientific data from rodents to humans [19,20,21,22,23,24]. Additionally, it is important to ensure that stem cell transplantation therapy is viable, through a confirmation of clinical and immunological safety as the essential factors in this process [25,26,27].

However, large animal experiments have several limitations, such as larger facility spaces and operating systems with specific pathogen-free conditions. In allogeneic transplantation, accommodation of the MHC between a donor and recipient remains difficult to resolve [28]. In this regard, our group has developed micro-mini pigs (MMPs) as large experimental animals to evaluate their safety and therapeutic validity in a preclinical study, and has already achieved a small colony by the technology of inbred lines and cloning [29,30]. Therefore, for the MHC problem, it has an increased chance of simulating the human family conditions as closely as possible.

As the background above, we selected the establishment of the gap nerve injury therapy, because gap nerve injury is a severe and refractory injury in large animals and humans. In addition, through the mouse sciatic nerve gap injury model, the mouse and human Sk-34 cells showed quite higher nerve reconstitution capacity (axonal growth and formation of perineurium/endoneurium) and functional recovery (dominant muscle contractile function) [16]. Therefore, we think that the particular nerve regeneration capacity of Sk-34 cells should be applied as a clinical therapy [31].

Additionally, immunological response is also an important aspect following somatic stem cell transplantation. To clarify the immunological response of Sk-34 cells in peripheral nerve therapy, class I haplotypes of swine leukocyte antigen (SLA) were analyzed, making it possible to adjust transplantation to 50% or 100% correspondence to simulate the human family case. Basically, Sk-MSC transplantation is an autologous therapeutic strategy. However, if Sk-34 cells show a low rejection response and do not consider SLA correspondence (100%), the therapeutic chance will be enhanced for sharing among family members.

## 2. Materials and Methods

### 2.1. Animal Usage

For the large animal experiments, we used MMP produced by the Swine and Poultry Research Center, Shizuoka Prefectural Research Institute of the Animal Industry (Kikugawa 439-0037, Shizuoka, Japan). Green fluorescent protein transgenic (GFP-Tg) MMPs (12–14 weeks old, *n* = 4) were used as donors, and non-Tg MMPs (12–14-week-old, *n* = 16) were used as recipients. The GFP-Tg MMPs were prepared via the backcross method [32] based on GFP-Tg Jinhua pigs [33] and MMP [34,35,36].

Parent MMPs (one father and two mothers) were fixed as the basic concept of the present study (see below). The parent pigs were maintained in a small population. They were then selected using cloning technology with a constant genetic background (coat-color genotypes and microsatellite markers of SLA) [37] and kept heteropositive for GFP by natural mating [32]. All the study protocols were approved by the Tokai University School of Medicine Committee on Animal Care and Use (#210421).

### 2.2. Determination of SLA Class I Haplotypes

The SLA haplotypes were determined as follows: whole blood cells from all animals used in the transplantation experiment (GFP-Tg and non-TG MMP) were treated with an RNA isolation reagent (TRIzol, Thermo Fisher Scientific Inc., Tokyo, Japan). Ear biopsy tissues were obtained from male and female parent animals, and prepared for the analysis. For the offspring, blood samples were used. Total RNA was isolated directly from the TRIzol-treated samples, according to the manufactural protocol. Complementary DNA (cDNA) was synthesized using oligo d(T) primers and the ReverTra Ace reverse transcriptase reaction (TOYOBO CO. Ltd., Osaka, Japan) after treatment of the isolated RNA with DNase I (Thermo Fisher Scientific Inc., Tokyo, Japan). A previously designed single primer pair (SLA-F1/F2 and SLA-R1/R2) that amplifies all known class I alleles was used for next-generation sequencing (NGS) [38]. Reverse transcription polymerase chain reaction, cycling parameter, NGS using Ion Torrent PGM system (Thermo Fisher Scientific Inc., Tokyo, Japan), and data processing were performed according to the previously published methods [38,39]. The SLA class I alleles were assigned by matching the trimmed and barcode-labeled sequence reads with 99% and 100% matching, 150 minimum overlap length, and 10 alignment identity score parameters, with all known SLA class I allele sequences released in the IPD-MHC database [40] using GS Reference Mapper Ver. 3.0 (Roche Diagnostics, Tokyo, Japan).

### 2.3. Nerve Gap Injury Model and Cell Transplantation

Nerve gap injury was induced in the left hind limb peroneal nerve based on previous mouse and rat studies [16,41]. Operation was performed under inhalation anesthesia (Sevoflurane; Nikko, Gifu, Japan) after anesthesia induction using methoxamine hydrochloride 0.08 mg/midazolam 0.08 mg per kilogram body weight (i.m.). Body temperature was monitored using a rectal probe and maintained at 36 ± 1 °C with radiant heat throughout the surgical procedure. During the surgery, an analgesic nonnarcotic opioid (butorphanol tartrate; 0.1 mg/kg subcutaneous infusion, Meiji Seika Pharma, Tokyo, Japan) was administered as needed. The left common peroneal nerve was exposed in the lower part of the thigh and completely transected (Figure 1A,B). Both transected stumps were bridged using a medical collagen tube (23–25 mm long, diameter 2.0–2.3 mm, Renerve, NIPRO, Osaka, Japan) with 1.5–2.0 mm fixative margin in both proximal and distal nerve stums (Figure 1B,C). Thus, the length of the nerve gap was adjusted to 20 mm or more.

The donor cells, which were isolated from GFP-Tg MMP skeletal muscle and stored in liquid nitrogen, were thawed and sorted CD34+/CD45− (Sk-34) cells. Then, Sk-34 cells were expanded by the method previously reported for 10–14 days [17]. Thereafter, Sk-34 cells were suspended in the Dulbecco’s Modified Eagle Medium [17] and were injected into collagen tubes concentration of 1 × 10^7^ cells/100 μL/nerve to fill the tube (Figure 1D,E). After the surgery, the surgical wound was sutured, and a transparent sterile/analgesic plastic dressing (Nobecutan spray; Yoshitomi Chemical, Tokyo, Japan) was applied over the wound. Penicillin (4000 units/100 mL) was administered subcutaneously to prevent infection. Buprenorphine 5–20 µg/kg was administered at 6–12 h intervals (i.m.) after the surgery as necessary.

### 2.4. Functional Assessment

Functional assessment was performed based on daily video observations of 15 min of sedentary behavior before, during, and after feeding. The pigs were reared in individual breeding cages (150 × 90 cm). Evaluation/scoring was performed by three special growers who were unaware of the transplantation conditions at each MMP. Inductive feeding was also used to move around the cage space, and the score was confirmed several times. For scoring, the fourth grade was assigned to recover points based on the appearance or disappearance of operated hind limb paralysis (such as sprained ankle/coffin joint; Figure 2). The detailed standards are listed in Table 1. Scoring was based on the point-deduction scoring system, and Score 3 was the highest recovery (no point reduction).

### 2.5. Immunohistochemistry

In vivo therapeutic potential of the cell transplantation was examined using immunohistochemistry. After 5–8 weeks, the recipient MMPs (*n* = 16) were anesthetized with an overdose of pentobarbital (100–120 mg/kg, intraperitoneally), and deep sleep was induced. After blood sampling and removal, the operated peroneal nerve and tibialis anterior (TA) and extensor digitorum longus (EDL) muscles in both the hind legs were sampled. The muscle samples were weighed, and the atrophy (decreasing) ratio against the right control side was calculated as the dominant muscle affected by the peroneal nerve lesion, and the results were compared with the functional recovery.

The operated nerves were removed with tubes (if remaining) and fixed with 4% paraformaldehyde/0.05 M phosphate buffer saline (PBS) overnight in situ. The samples were then washed with 0.01 M PBS, followed by treatment with a graded 5–25% sucrose/0.01 M PBS series to dehydrate, embedded in OCT compound, and frozen/stored at −80 °C until use.

For staining of the histological sections, several 7 µm cross-sections of the operated peroneal nerves were obtained. The nerve axons were detected using N200 (1:500, anti-neurofilament 200 monoclonal Clone NE14, N5389, Sigma, St. Louis, MO, USA). Anti-glucose transporter 1 (1:500, rabbit polyclonal antibody; Diagnostic Biosystems, Pleasanton, RP128, CA, USA) was used to detect the perineurium/endoneurium. Furthermore, blood vessels were detected using mouse monoclonal α-smooth muscle actin (αSMA, Clone 1A4, Cy3-conjugated directly; 1:1500, for 1 h at room temperature; C6198, Sigma, St. Louis, MO, USA). The reactions were visualized using Alexa Fluor-488- and 594-conjugated goat anti-rabbit and anti-rat antibodies (1:500 dilution; Molecular Probes, Eugene, OR, USA) for 2 h at room temperature. The nuclei were counterstained with 4′,6-diamino-2-phenylindole. Engrafted GFP-Tg pig cells were enhanced using rabbit anti-GFP IgG fraction (1:300, A11122; Molecular Probes, Inc., Eugene, OR, USA).

### 2.6. The Summary of This Study

All experimental protocols are summarized in Figure 3.

As a limitation of this study, the large animal surgery room can only be conducted with two animals per day at our facility maximum. Additionally, the pig and goat breeding facility could be used for 5 months by rotation, both limited to 4 animals at once. This is the reason for the differences in experimental evaluation periods.

### 2.7. Statistics

Statistical analysis for Figure 4 and Figure 5 was performed using the parametric Tukey–Kramer post-hoc test, and the significance level was set at *p* < 0.05. Values are expressed as mean ± standard error.

## 3. Results

### 3.1. Individual Changes in Body Weight

Individual changes in body weights of all the operated animals are shown in Figure 3. All the animals showed favorable growth curves during the experimental period. The term includes breeding at the Swine and Poultry Research Center, as well as transportation to our university facilities. After 5–10 weeks of comfortable breeding in individual normal cage environments, the nerve gap surgery was performed, and the animals were returned to the same environment.

### 3.2. Class I SLA Haplotypes

The SLA class I haplotypes of the parent MMPs and their offspring are summarized in Figure 5. A homozygous SLA holotype was present in both mother animals. Furthermore, a father of the MMP had an identical haplotype in one genetic locus. Thus, the offspring in the present study were determined to have two types of SLA haplotypes. Therefore, we selected two types of offspring with GFP-Tg-positive MMP as donors and non-GFP-Tg offspring as recipients. Therefore, all transplantations showed 50% and/or 100% matches with the SLA class I haplotypes.

### 3.3. Functional Recoveries

The functional recovery score based on the general behavior and weight decrease (atrophy) ratio of the peroneal nerve dominant muscle is shown in Figure 6A,B. The transplantation groups showed rapid recovery within the first two weeks, and then achieved no abnormalities (score 3) within three weeks. However, a non-cell-transplantation group (collagen tube bridge only) did not achieve a recovery score of 3, even after the extension of the recovery period by 3–5 weeks compared to the other two groups; this may be a big difference. With regard to the SLA correspondence, the 100% group showed a statistically significant faster recovery period for a score of 1 than the 50% group. However, this period may not be clinically important (Figure 6A).

No differences were observed in the muscle atrophy ratios of the TA and EDL of the three groups. All three groups experienced uniform muscle atrophy due to denervation (decreased wet weight; Figure 6B).

### 3.4. Histochemical Evaluation of Functional Recovery

Histological recovery of each operated nerve was analyzed using immunohistochemistry (Figure 7). Sections were consistently obtained from the distal portion of the bridged tubes, which was a short (not reached) distal nerve stump (Figure 7A). The Sk-34-transplanted nerve showed favorable recovery of nerve fibers, with a good number of axons associated with the perineurium/endoneurium (Figure 7B–D). Additionally, the formation of several blood vessels was observed (Figure 7E). In contrast, axon formation was scarce and sparse in the non-cell transplantation nerve (Figure 7F–H), as well as the blood vessels (Figure 7I). All 4 animals of the non-cell-transplanted group showed a similar trend, and this trend well represented functional recovery. No inflammatory reactions were detected in the two Sk-34 cell-transplanted groups.

## 4. Discussion

The current study found that feeding/living environmental stress was low or almost zero due to the positive body weight change without large decreases in all animals, as confirmed in Figure 4. There was no decreased feeding behavior reported from the breeding staff through the experimental period.

Fortunately, the current study successfully obtained only two SLA haplotypes for offspring, as expected, from one paternal and two maternal MMPs (Figure 5). We achieved Sk-34 cell transplantation with 50% or 100% SLA correspondence.

Regarding functional recovery, it was assumed that the change from a recovery score of 0 to 1 was mostly attributed to a decreased uncomfortable feeling of instant pain. Consequently, this was thought to be a type of adaptation (or habituation) and did not reflect the recovery of constitutive nerves. Thus, the clinical significance of a score change of 0–1 was not significant. However, score changes from 2 to 3 are considered very important in the clinical phase of nerve recovery because it was considered that during 2–3 weeks after, the nerve regeneration phase is largely affected by the cell transplantation effects of Sk-MSCs, as shown in mice and human studies [16,41].

The differentiation of Schwann and perineurial/endoneurial cells, which are associated with vascular cells (pericytes, vascular smooth muscle, and endothelial cells), have been demonstrated in the sciatic nerve of rodents [41]. Additionally, the expression of paracrine factors in the transplanted Sk-MSCs was reported to be effective in nerve regeneration [16,18].

The present large animal experiment also showed apparent differences in functional recovery levels between the non-cell-transplanted and Sk-34-transplanted groups (Figure 6A). Therefore, it is reasonable to conclude that transplantation of Sk-34 cells is clinically effective in the three species. The functional recovery score of the present study between 2 and 3 may be a critical difference in nerve gap injury treatment, depending on whether Sk-34 cells are used or not.

In contrast, the functional recovery scores matching the SLA of the 50% and 100% groups were similar (Figure 6A). Thus, it can be assumed that this difference is negligibly small, and that Sk-34 cell transplantation for nerve lesions may be sharable among the family in humans. Additionally, it was speculated that the immunological rejection response to Sk-34 was low and negligible in allogeneic transplantation, which would make this therapy more attractive. The human Sk-34 and Sk-DN cells could be obtained from both the legs and abdominal muscles [15]. Therefore, a small sample removal (around 3g) from the abdominal wall muscle would allow sufficient donor cells relatively easily and safely, probably with lower impact on the body than the appendiceal operation, because the bowels are untouched.

The aforementioned functional recovery was supported by immunohistochemical analysis (Figure 7). Favorable nerve fiber (axon) regeneration with the formation of perineurium/endoneurium and blood vessels was consistently observed in the Sk-34 cell-transplantation group (50 and 100%) in the distal portion of the bridged tube, whereas the non-cell-transplantation group showed scarce and sparse axonal formation (Figure 7). This trend corresponded to the functional recovery score (Figure 6A).

Regarding the recovery period, the current recovery terms of 5–8 weeks are shorter than the previous rodent reports using Sk-34 cells, which were conducted during 8–12 weeks. For this reason, the sciatic nerve that is closer to the central nervous system (spinal cord) was used in the rodent model, and the present peroneal nerve just close to the knee joint is situated more peripheral. This may be the possible reason for a short time recovery of the current MMP model.

In regard to using the GFP-Tg MMP as the donor animal in the present study, we could not detect the positive GFP reactions in the recipient MMP nerve, even after enhancement by anti-GFP application. For this reason, it is conceivable that the GFP strength of the donor cells was insufficient because of the difficulty of normal birth of GFP-Tg MMP by homogenous mating. Therefore, it could withstand heterogeneous mating, but the GFP strength decreased. The relationship between the GFP emission strength and its reduction in the engrafted recipient tissue was an injected tissue volume dependence, and this was reported previously [17].

The results of functional recovery did not always show muscle atrophy (Figure 6), so we assumed that the living environment of the MMP was significantly affected by these results. In our facility, the pigs could not leave their individual cages in a wider space, such as an exercise room; thus, they could not walk or run. Muscle weight recovery may be affected by limited exercise in a limited space for all animals, which may explain the lack of difference. However, denervation is a serious event for the motor system [42,43], and if not reinnervated, the downstream muscles might develop cord-like features. If the study terms had been extended and exercise had been available, the results could have been altered.

The present study demonstrates the safety, validity, and availability of Sk-34 cell transplantation for nerve gap injury in large animals. Additionally, the facilitating effect of nerve regeneration and immunotolerance of Sk-34 cells was also demonstrated under 50% and 100% matching conditions of the SLA class I haplotypes. This suggests that donor Sk-34 cells are shared among siblings and parents. Therefore, this is the first and most significant advanced study using large animal somatic stem cells and attractive cell transplantation therapy in preclinical experiments [44]. Clinical trials are the possible next step in this direction.

## 5. Conclusions

The present study showed that skeletal muscle-derived Sk-34 cell transplantation accelerates the functional recovery of peripheral nerve gap injury, even under 50% matching of the SLA haplotypes as well as 100%. Therefore, we believe that Sk-34 cell transplantation therapy for nerve lesions should be applied clinically as the next step, or at least ready for use in veterinary medicine.

## Figures and Tables

**Figure 1 biomolecules-14-00939-f001:**
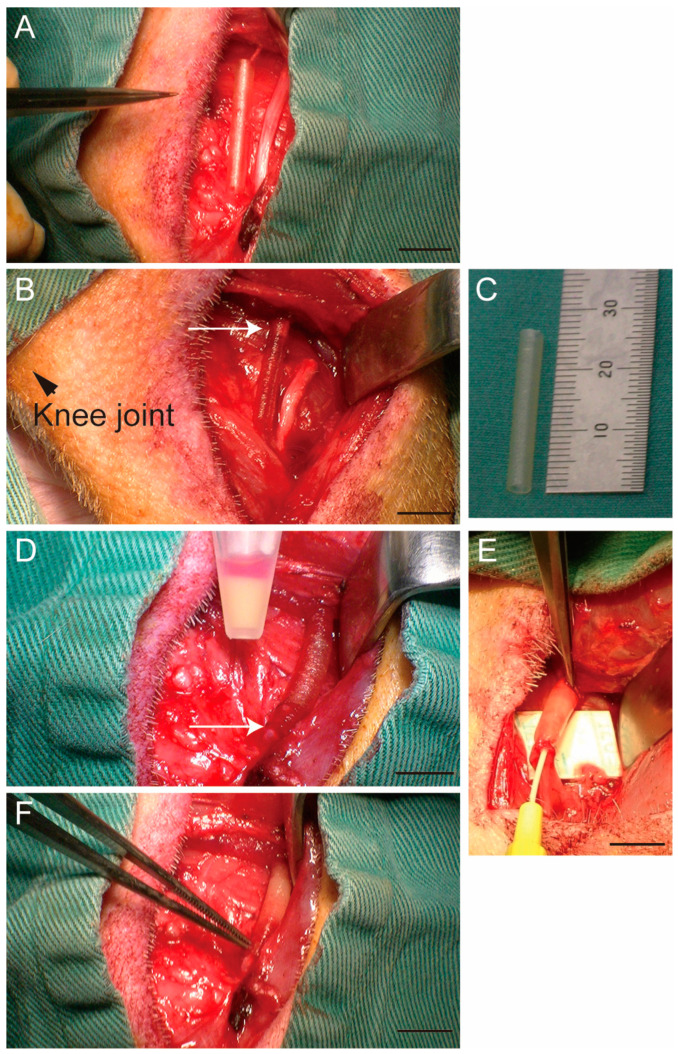
Preparation of nerve gap model (**A**–**C**), tube bridging, and cell transplantation (**D**–**F**). Suspended Sk-34 cells in microtube (**D**) were inserted into the bridging tube using 23G catheter and filled (**E**). Final state (**F**). Arrow in (**B**) shows the proximal stump of the peroneal nerve. Arrow in (**D**) shows the distal nerve stump. Bars = 10 mm.

**Figure 2 biomolecules-14-00939-f002:**
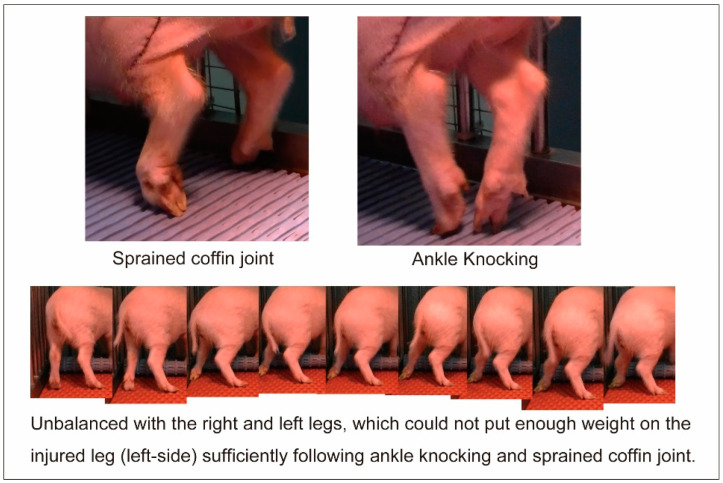
Typical behavior of operated hind limb paralysis, such as sprained ankle/coffin joint. The frequency of the appearance of these behaviors was reflected in the functional recovery score. Appendix A can be found on the website.

**Figure 3 biomolecules-14-00939-f003:**
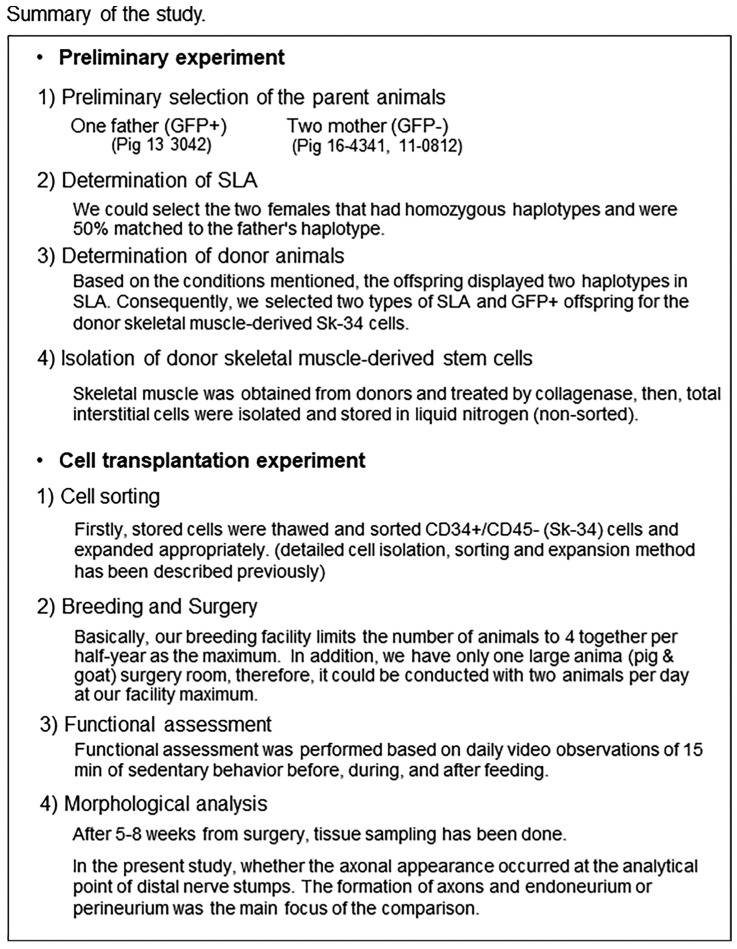
Summary of this study. Detailed cell isolation, sorting and expansion method has been described previously [17].

**Figure 4 biomolecules-14-00939-f004:**
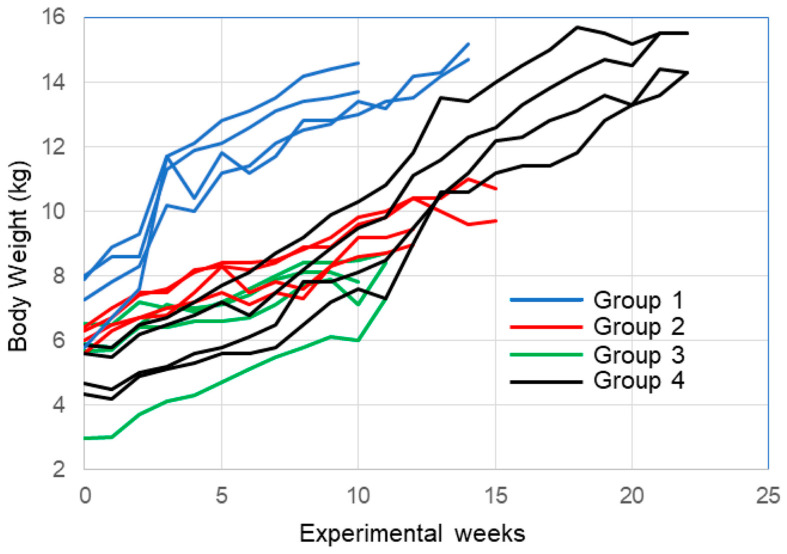
Changes in body weight of all operated animals. All the animals grew favorably in the experimental conditions. To be comfortable with the environment of our university facilities after transportation from the Swine and Poultry Research Center, 5–10 weeks of individual normal cage breeding was performed. Group 4 is a non-cell-transplanted group and 1–3 are cell-transplanted groups. The reason for the differences in experimental evaluation periods is that the large animal surgery room can only be conducted with two animals per day at our facility maximum. Additionally, the pig and goat breeding facility could be used for 5 months by rotation, both limited to 4 animals at once.

**Figure 5 biomolecules-14-00939-f005:**
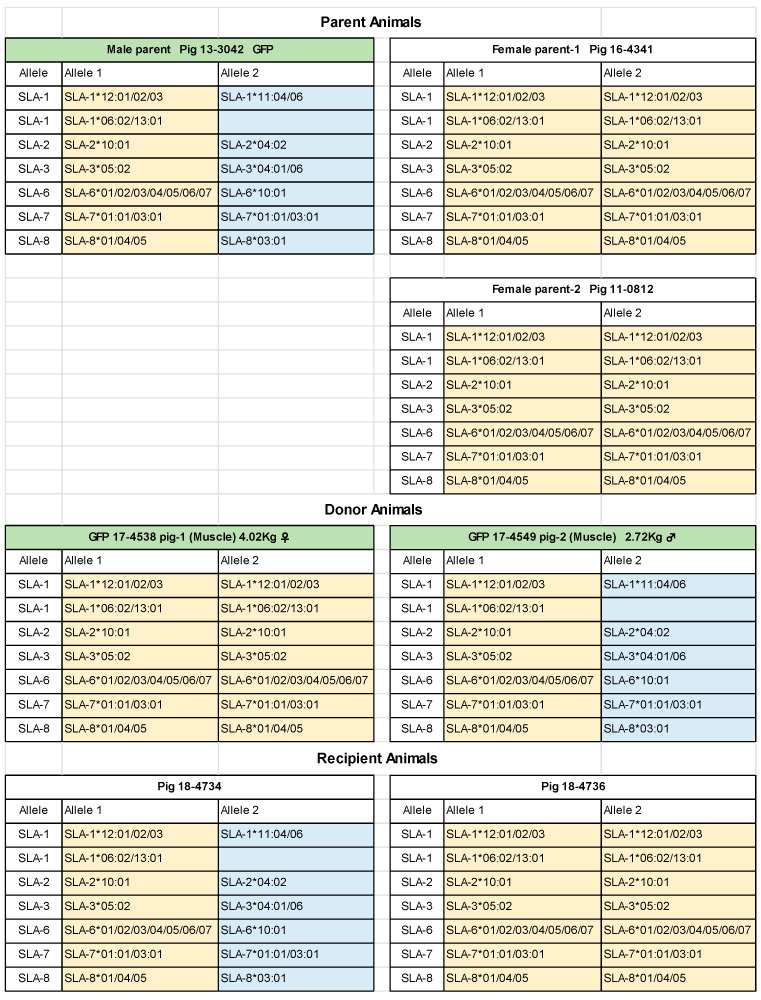
Swine leukocyte antigen class I haplotypes of parents and their offspring (donors and recipients). The two female parents had a homozygous haplotype and were 50% matched to the male parent MMP. As a result, the haplotype of the offspring is limited to two types. Top green color shows positive GFP-Tg. SLA, swine leukocyte antigen; GFP-Tg, green fluorescent protein transgenic.

**Figure 6 biomolecules-14-00939-f006:**
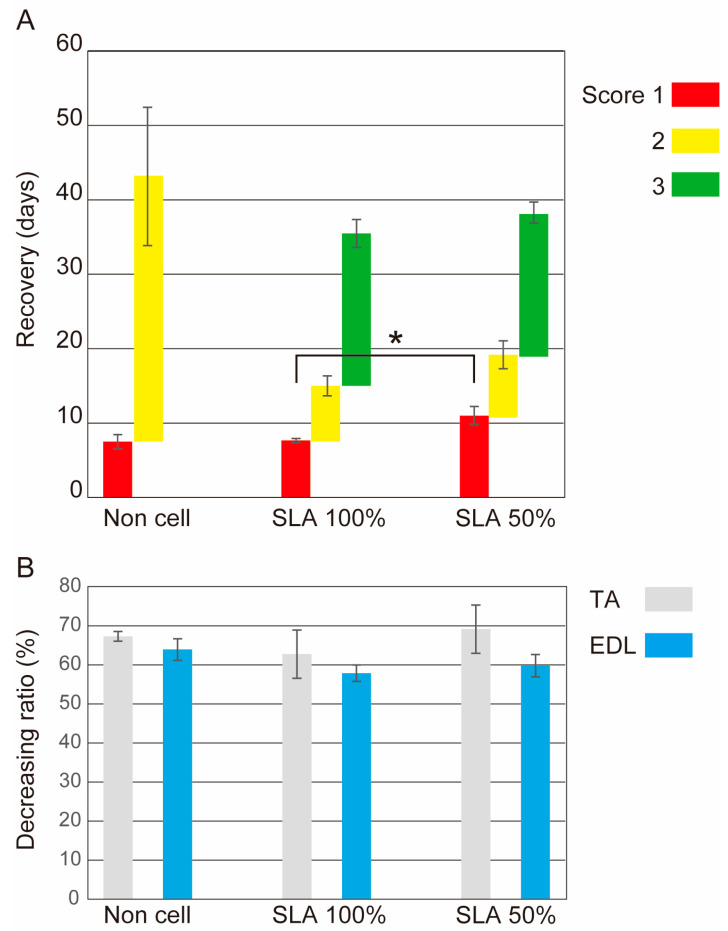
Functional recovery of the operated leg (**A**) and weight loss (atrophy) ratio of the TA and EDL muscles (**B**). Note that the non-cell-transplantation group failed to achieve a score of 3, even with a recovery period of 3–5 weeks longer than that in the transplanted groups. * *p* < 0.05. SLA, swine leukocyte antigen; TA, tibialis anterior; EDL, extensor digitorum longus.

**Figure 7 biomolecules-14-00939-f007:**
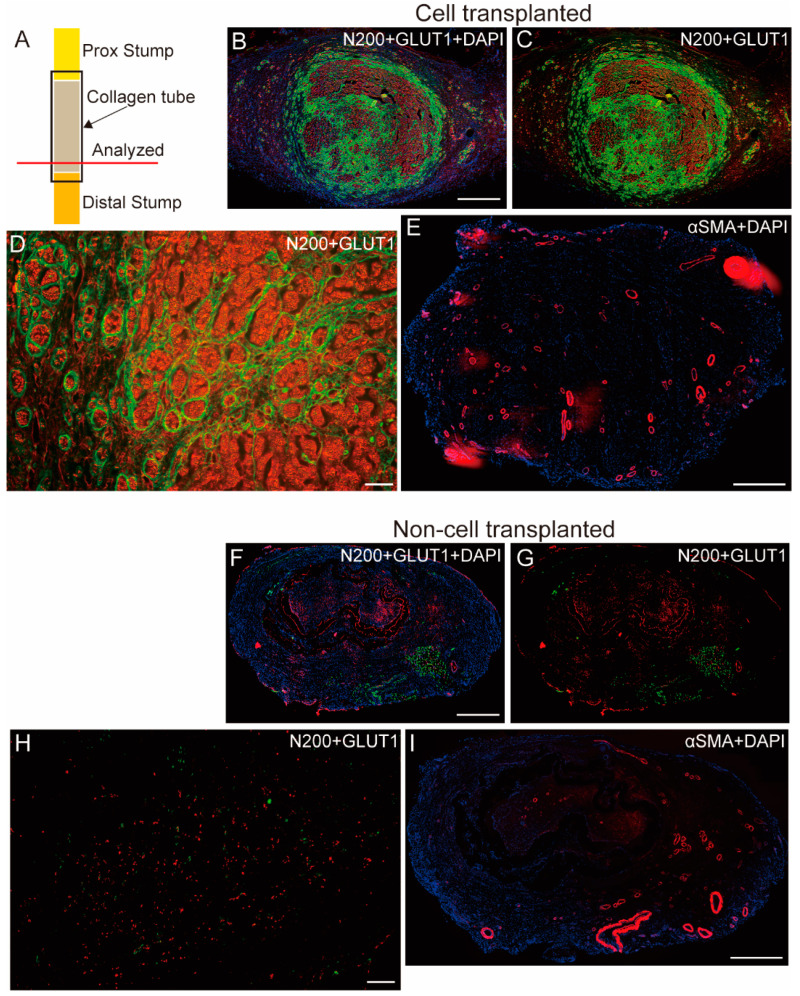
Immunohistochemical detection of nerve fibers, perineurium/endoneurium, and blood vessels in the bridged tubes. Panels (**A**–**E**) are from a Sk-34 cell-transplanted nerve, and (**F**–**I**) are from a non-transplanted nerve. (**A**): a depiction of tube bridging and the area that was examined by section. (**B**,**C**): whole sectional feature of the Sk-34 cell-transplanted nerve. (**D**): higher magnification of (**C**). (**E**): Blood vessel formation of the transplanted nerve. (**F**,**G**): whole features of non-cell-transplanted nerve. (**H**): higher magnification of (**G**). (**I**): blood vessel formation. Nerve fibers were detected by N200 (red), perineurium/endoneurium by glucose transporter 1 (GLUT 1, green), and blood vessels by α-smooth muscle actin fibers (αSMA, red). Nuclear staining was performed with 4′,6-diamino-2-phenylindole (DAPI) (blue). Bars in (**B**,**E**,**F**,**I**) = 1 mm, and (**D**,**H**) = 50 µm. Appendix A are available on the website.

**Table 1 biomolecules-14-00939-t001:** Functional recovery score sheet.

Score	Basis for Evaluation of Functional Recovery
Score 0	(1) Pig barely moves the weight on the operated leg with a slight shiver. (2) Clearly different fetlock height between right and left with unbalance (could not put weight on an injured leg sufficiently). (3) Contraction of antagonist muscles cannot work together and unfix the joint. (4) Frequently observed sprained coffin joint and knee knocking (>5 times, refer to Figure 2).
Score 1	(1) The trend above was relieved, such as the fetlock height being almost the same, and the sprained unguis was reduced several times (2–5 times). (2) Shivering decreased when shifting weight.
Score 2	(1) Sprained coffin joint decreased within 1–2 times. (2) Shivering disappeared. (3) Behavioral activity increased.
Score 3	(1) Above abnormality was not observed completely. (2) Looks healthy and active.

## Data Availability

The original contributions presented in the study are included in the article/Appendix A, further inquiries can be directed to the corresponding author.

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
