# Peer review of "Skeletal Muscle-Derived Stem Cell Transplantation Accelerates the Recovery of Peripheral Nerve Gap Injury under 50% and 100% Allogeneic Compatibility with the Swine Leucocyte Antigen"

_biomolecules, 2024, doi:10.3390/biom14080939_

Round 1

Reviewer 1 Report

Comments and Suggestions for Authors

Tamaki and colleagues presented an original work demonstrating the successful use of skeletal muscle-derived stem cells to recover peripheral nerve gap injuries under 50% and 100% allogeneic compatibility with swine leucocyte antigen.  To improve the quality of the paper, some suggestions are shown below:

A) Line 60: Sentence not completed.

B) Line 73: Bibliographic citation failure.

C) Table 1 and Figure 2: The inclusion in supplementary materials of representative videos demonstrating the animals' behavior during post-surgical recovery score assessments would be quite enriching for the work.

D) Lines 189-200: Please include the product code for each antibody used.

E) Section 2.5: Describe in the methodology how immunofluorescences were analyzed.

F) Materials and Methods section: I strongly suggest the inclusion of a schematic figure, such as those made in biorender, briefly illustrating the work methodology. In other words, from obtaining the stem cells to the scheme of their infusion in animals to observe the regeneration of the lesion.

G) Figure 3: It is unclear why some animals had different experimental evaluation periods. Also, include the justification in the figure legend.

H) Figure 3: The different colors of each line referring to the weight of the animals could be separated by experimental group.

I) Lines 271-273: The commented results are not presented in the text. I suggest its inclusion.

J) Lines 277-302: It's a very long paragraph, please break it into smaller ones.

Author Response

Reviewer 1

Thank you for your kind review of our manuscript.

Changed text was colored red for easy identification.

  1. Line 60: Sentence not completed.

Response: Thank you for the kind suggestion, and we have completed sentence L 60. 

  1. Line 73: Bibliographic citation failure.

Response:    We really appreciate the reviewer for noticing such important things. We performed a re-scan based on the end note.

  1. Table 1 and Figure 2: The inclusion in supplementary materials of representative videos demonstrating the animals' behavior during post-surgical recovery score assessments would be quite enriching for the work.

Response:  We have passed over this issue because of the animal protection. However, we are sure that this is the right thing. Therefore, we decided to present the video as a supplemental material. We believe that this video might be a signal for the truth of this study.      

  1. Lines 189-200: Please include the product code for each antibody used.

Response:  we include the product code for each antibody.   

  1. E) Section 2.5: Describe in the methodology how immunofluorescences were analyzed.

Response:  Thank you for your suggestion. This has been included in new Figure 3, as mentioned below.

  1. F) Materials and Methods section: I strongly suggest the inclusion of a schematic figure, such as those made in biorender, briefly illustrating the work methodology. In other words, from obtaining the stem cells to the scheme of their infusion in animals to observe the regeneration of the lesion.

Response: The reviewer is right. Several divisions were involved in conducting the current study. Therefore, we inserted one more figure (new Figure 3) as the “Summary of this study”. This may be a helpful for the understanding of methodology. In addition, we would like to read our previous study [17].

  1. G) Figure 3: It is unclear why some animals had different experimental evaluation periods. Also, include the justification in the figure legend.
  2. H) Figure 3: The different colors of each line referring to the weight of the animals could be separated by experimental group.

Responses for G and H: Firstly, we enlarged Figure 3, then, colors of each line were separated by experimental group.  Also, a reason of the different experimental evaluation periods is included in the figure legend. This difference did not affect the functional recovery period (Figure 5).  The above, new Figure 3 “Summary of the study” also useful to understand the individual differences.

  1. Lines 271-273: The commented results are not presented in the text. I suggest its inclusion.

Response: The following is included in the text about this matter.

The current study found that feeding/living environmental stress was low or almost zero due to the positive body weight change without large decreases in all animals, as confirmed in Figure 3. In particular, the last 5-8 weeks were actual recovery period.  There was no decreased feeding behavior reported from the breeding staff through the experimental period.

  1. J) Lines 277-302: It's a very long paragraph, please break it into smaller ones.

Response: Yes, the reviewer is right, there were confusions. The paragraph was broken several times.

Reviewer 2 Report

Comments and Suggestions for Authors

The authors assess the effect of skeletal muscle derived stem cell transplantation in a nerve gap injury micro-mini-pig model under 50% and 100% allogeneic SLA compatibility. The study is well performed generally but the authors should address the following concerns,

1. The individual body weight data shown in Figure 3 is not complete for all animals as the weights are missing for some animals in the later time points. How does the body weight differ between the different treatment groups over time?

2. The Fig 5A shows the scoring of the functional recovery over time in the different groups. Although the improvement in score is evident between cell transplanted and non-transplanted groups, it would be good if the authors could add some pictures of the pigs in different groups if available as shown in figure 2. Also, the authors should show individual data points and add error bars to all the bars.

3. Figure 6 shows clear recovery of nerve fibers with axons associated with peri/endoneurium in the cell transplanted group. However, the images are shown just from one animal. How consistent is the recovery pattern across animals in the cell transplanted and non-transplanted groups? 

4. The authors mention that they could not detect GFP even after antibody staining and they indicate that this is due to weak expression. The authors should clarify how they can conclude that it is not due to a technical issue. 

Comments on the Quality of English Language

English language quality is fine. No major issues detected but minor editing is recommended. 

Author Response

Reviewer 2

Thank you for your kind review of our manuscript.

Changed text was colored red for easy identification.

  1. The individual body weight data shown in Figure 3 is not complete for all animals as the weights are missing for some animals in the later time points. How does the body weight differ between the different treatment groups over time?

Response: Thank you for your suggestion, and we aware that we need further explanation. Therefore, we added the new Figure 3 as “Summary of the present study”. This may be helpful for the understand these differences. The large animal experiments mostly require an extensive system and team members.

  1. The Fig 5A shows the scoring of the functional recovery over time in the different groups. Although the improvement in score is evident between cell transplanted and non-transplanted groups, it would be good if the authors could add some pictures of the pigs in different groups if available as shown in figure 2. Also, the authors should show individual data points and add error bars to all the bars.

Response: In this revision, we added one video based on scoring. Therefore, could you please image the functional differences? It's hard to tell the difference from the quiescent picture, but the immunohistochemical data (Figure 5) clearly showed them. Functional assessment was performed daily, and the number of individual points was huge. Error bars of score 3 in both 50 and 100% groups were missed, and added in revision.

  1. Figure 6 shows clear recovery of nerve fibers with axons associated with peri/endoneurium in the cell transplanted group. However, the images are shown just from one animal. How consistent is the recovery pattern across animals in the cell transplanted and non-transplanted groups? 

Response: The reviewer is right, and this was confusable, but all 4 animals of non-cell transplanted group showed a similar trend and this trend well represented functional recovery. We added this explanation in the results.

  1. The authors mention that they could not detect GFP even after antibody staining and they indicate that this is due to weak expression. The authors should clarify how they can conclude that it is not due to a technical issue. 

Response: We also worry about this issue and we have reported in detail [17]. The same issue also experienced by the case of GFP-rat.

Round 2

Reviewer 2 Report

Comments and Suggestions for Authors

The authors have addressed many of the concerns raised in the first round of review. In lines 270-271 of the revised manuscript the authors added a statement saying all the animals belonging to the cell transplanted group showed the same staining pattern for the markers assessed in Figure 7. But it would be good to include data to support this statement. For this, including data on quantification of the staining of N-200, GLUT1 and a-SMA in different animals would be good. The authors could also include the staining images for the remaining animals in the supplementary section for completeness. 

Comments on the Quality of English Language

English language quality is fine but spell check and grammar check is recommended.

Author Response

The authors have addressed many of the concerns raised in the first round of review. In lines 270-271 of the revised manuscript the authors added a statement saying all the animals belonging to the cell transplanted group showed the same staining pattern for the markers assessed in Figure 7. But it would be good to include data to support this statement. For this, including data on quantification of the staining of N-200, GLUT1 and a-SMA in different animals would be good. The authors could also include the staining images for the remaining animals in the supplementary section for completeness. 

Response:

The reviewer is right. Quantification for histological analysis is very important, and we have done it in the previous mice and rats sciatic nerve experiments. Then, we achieved over 10 works for the publication. That is way, we tried to quantify several methods, however, the size of nerve 6-10 times larger in pig peroneal nerve than mice/rat sciatic nerve, and redistribution of peripheral nerve fascicle were well developed in pig. Also, It was hard to judge whole nerve fiber fascicle (covered each other by perineurium).  Furthermore, the number fibers also higher in pigs about 10 times or more. The number of nerve fibers in the sciatic nerve of mice and rats ranges between 4 and 6 thousand, but it is possible for pigs to have up to 40-60 thousand or more. The number of nerve fibers in the sciatic nerve of mice and rats ranges between 4 and 6 thousand, but it is possible for pigs to have up to 40-60 thousand or more. Analyzing at the light microscopic level was a challenge, so we gave up counting. In fact, it's impossible, and if we tried, the accuracy would be reduced. However, in terms of results, immunohistochemical features of 2 transplanted and non-cell transplanted group was very clear in one look. We used conventional visual effects to conduct a comparison, and these results accurately reflected functional recovery.  

However, we would like to understand that preparing the entire sciatic nerve feature is not so easy because it requires tiling multiple photographs. For example, Figure 7 B is a tiling image that we have also used in mice and rats sciatic nerve. Actually, for mice/rat cases, 4-8 photographs (20x20) were needed for tiling to obtaining whole nerve feature. But in pigs, 20-50 photographs are necessary. In such case high spec PC with high grade processor (intel core i7) and over 32 GB memory must be needed. Due to the difficulty of comparing the 50 and 100% groups, a conventional visual image was employed to determine that the results were not significantly different. This is obvious from the functional recovery (Figure 6). The above are the reasons for the presentation of the current study.  Actually, reports using whole fiber feature (even in the mice/rat) is rare case.

Our addition was as follows. (273-274).

All 4 animals of non-cell transplanted group showed a similar trend and this trend well represented functional recovery.

Thus, please check again. It is likely that in a backward way, we would upload the pictures to other animals. However, we added supplemental figures.